# Hardware Efficient Solutions for Wireless Air Pollution Sensors Dedicated to Dense Urban Areas

**Marzena Banach** [1,*], **Rafał Długosz** [2,*], **Jolanta Pauk** [3] **and Tomasz Talaśka** [2]

1   Institute of Architecture and Spatial Planning, Poznan University of Technology, Nieszawska 13C,
    61-021 Poznań, Poland
2   Faculty of Telecommunication, Computer Science and Electrical Engineering, UTP University of Science and
    Technology, 85-796 Bydgoszcz, Poland; talaska@utp.edu.pl
3   Institute of Biomedical Engineering, Faculty of Mechanical Engineering, University of Technology,
    Wiejska 45A, 15-351 Bialystok, Poland; j.pauk@pb.edu.pl
*   Correspondence: marzena.banach@put.poznan.pl (M.B.); rafal.dlugosz@utp.edu.pl (R.D.);
    Tel.: +48-668-160-217 (R.D.)

**Abstract:** This paper proposes novel solutions for the application of air pollution monitoring systems in so called 'smart cities'. A possibility of the implementation of a relatively dense network of wireless air pollution sensors that can collect and process data in real time was the motive behind our research and investigations. We discuss the concept of the wireless sensor network, taking into account the structure of the urban development in cities and we present a novel signal processing algorithm that may be used to control the communication scheme between particular sensors and an external network. We placed a special emphasis on the computational complexity to facilitate the implementation directly at the transistor level of particular sensors. The algorithm was verified using real data obtained from air pollution sensors installed in Krakow, Poland. To ensure sufficient robustness of the variability of input data, we artificially added high amplitude noise to the real data we obtained. This paper demonstrates the performance of the algorithm. This algorithm allows for the reduction of the noise amplitude by 23 dB and enables a reduction of the number of wireless communication sessions with a base station (BS) by 70%–80%. We also present selected measurement results of a prototype current-mode digital-to-analogue converter to be used in the sensors, for signal resolutions up to 7 bits.

**Keywords:** air pollution sensors; air pollution maps; edge computing; pollution data processing; analog-to-digital converters; low power circuits

## 1. Introduction

Various types of air pollution, especially those designated as particulate matter $PM_{2.5}$ and $PM_{10}$ (smog), are hazardous to human health [1] (according to the United States Environmental Protection Agency). This applies especially to those particles with a diameter below 2.5 µm, as their size allows them to enter the human blood directly. In the case of $PM_{10}$ particles (a diameter less than 10 µm) carcinogenic heavy metals, benzopyrenes furans, dioxins, etc., are the most harmful [2–4].

Air pollution is a risk for many diseases. Epidemiological evidence indicates that children are more susceptible to PM health effects than adults. In terms of long-term exposure, fine particles ($PM_{0.1}$–$PM_{2.5}$) represent a higher risk factor than coarse particles ($PM_{2.5}$–$PM_{10}$). In the study [5], an airborne particulate matter effect on young children's respiratory diseases was provided. They observed that exposure to airborne fine particulate matter ($PM_{2.5}$) carries substantial health risks, particularly for younger children (0–10 years). This is because children under 10 years have a faster breathing rate compared to adolescents and adults, which results in inhalation of higher quantities of air.

In [6], a significant impact of air pollution on diseases of the upper respiratory tract is presented. According to the authors, air pollution is associated with global warming.

According to the British Heart Foundation, airborne particulate matter (PM) can increase the risk of cardiovascular disease (CVD) in three ways [6,7]. First, PM can stimulate receptors in the lungs that disrupt the nervous system and cause changes to the heart rhythm. Secondly, inhaled particles can cause inflammation of the lungs and then damage the cardiovascular system as the inflammatory chemicals pass into the blood. Eventually ultrafine particles may pass into the blood and directly affect the blood vessels [7]. In the study [8], the effects of environmental air pollution on the progression of cardiovascular problems were studied. They stated that environmental pollution can cause high blood pressure, arrhythmias, enhanced coagulation, thrombosis, acute arterial vasoconstriction, atherosclerosis, ischemic heart diseases, myocardial infarction, and even heart failure. Despite existing norms of the levels for air pollution in each country, there is no evidence of a safe level for PM exposure or a threshold below which no adverse health effects may occur [5].

Taking into account the possible impact on health, the acceptable levels of $PM_{2.5}$ and $PM_{10}$ pollution are provided in the norms for particular countries and by the World Health Organization (WHO) [9–11]. Air pollution sensors and systems also have to comply with existing standards [12,13]. The standards may differ across the world. In Poland, for example, European standards apply.

The prevention of pollution is one of the fundamental ecological goals of theoreticians of so-called smart cities. This ecological aspect is closely related to city transport, as road transport is one of the primary sources of air pollution in urban areas. The dependence on transport affects the daily fluctuations of pollution levels in cities. For this reason, it is vital to develop new air monitoring systems, that can quickly inform residents about changes in the pollution levels. This will be of particular benefit to pedestrians and cyclists, the two groups who are the most exposed to air pollution when moving around the city.

To address this issue, one of the questions this paper focuses on is a novel, flexible air pollution monitoring system based on wireless sensors densely distributed in the urban environment—wireless sensor networks (WSN). The idea is to monitor the level of pollution in the streets of the city, where the maps of pollution distribution need to be updated regularly. Such a system would allow pedestrians and cyclists to plan their routes in the city to avoid the most polluted areas.

To achieve the described objectives, sensors used in the system should be cheap and easy to install in the urban infrastructure. One of the important concepts is to reduce the energy consumed by the WSN nodes so that they can operate using the energy sourced from the local environment. This, in turn, requires the optimization of particular components of the hardware layer of the sensors towards low power dissipation. In this paper, we focus on the development of such solutions. These include a novel specialized algorithm responsible for controlling the use of radio frequency (RF) data transmission, an analogue-to-digital converter (ADC), and filters. We devised and implemented most of these components as a specialized prototype chip (application-specific integrated circuit (ASIC)).

The investigations presented in this work form one part of a larger research project. The project aims to develop high spatial density air pollution maps for smart cities. This project is being conducted in an interdisciplinary team, involving several universities in Poland and several research disciplines (information technologies and electronics, architecture, and urban spatial planning, as well as biocybernetics).

The project has so far resulted in the development of hardware components of low power air pollution sensors, for example, specialized signal processing algorithms, analogue-to-digital (ADC) and digital-to-analogue converters (DAC), and filters, etc. Selected results obtained when testing the described components are discussed in this work. There are plans to build a fully functional sensor for verification in selected cities in Poland.

The paper is organized as follows. In the second part, the state-of-the-art systems of air pollution monitoring and the structure of selected pollution sensors are discussed. In the third section, the design details and a description of both the programming environment and the hardware are presented.

The fourth chapter is devoted to the hardware-efficient algorithm to detect changes in pollution levels. The following two sections present the test results and a discussion. The conclusions are included in the last part of the manuscript.

## 2. Related Works

### 2.1. Air Pollution Monitoring Systems

Conventional air pollution monitoring is based on large, typically expensive stations, with an average price at the level of 60,000 USD. Their number is relatively small. Consequently, the resolution of the resultant air pollution map is small too. In Poland, for example, stations of this kind are located, predominantly, in the most sensitive parts of the country, usually in larger cities [14]. The number of measuring stations amounts to 1924, with 1098 automated ones (57%). They provide information on the concentration of sulfur dioxide, carbon monoxide, ozone, nitrogen oxides, benzenes, and $PM_{10}$ and $PM_{2.5}$ particles. One of the crucial problems is the fact that the public only rarely has access to updated information.

In recent years, there has been a trend to develop more dense networks of air pollution sensors. One of interesting solutions of this type is offered by the Airly company in Poland. Relatively cheap sensors, with the price at the level of 300 USD, allow for building higher resolution air pollution maps. The measurement results are provided throughout the Airly company's web page [15]. An important feature of this system is a higher frequency of pollution map updates (every hour). The maps created by Airly show that even substantial differences between pollution levels may be observed in points that are close to each other.

Another recently reported solution is a concept of an air monitoring system based on mobile sensors mounted on buses (public transport) [16]. In [16] pollution measurements are carried out once every minute. In this case the measurements are carried out exclusively along transit lines—which is a disadvantage. Another disadvantage is that if no bus has been moving along a particular transit line for a longer time, the information on pollution levels in that area is not available. In this context, it is also worth mentioning the wearable pollution sensors that are already available on the market [17].

The system offered by Airly, as well as the one described in [16], motivated us to undertake investigations in this area. In the following parts of the paper, we contribute to the development of low power stationary sensors that could become the basis of a monitoring system offering higher resolution maps.

### 2.2. Air Pollution Sensors

From the technological side, the functioning of smart cities requires the use of various types of sensors, providing data in real time [18]. The key features of such devices include having a small size and low energy consumption [19]. This creates the possibility of implementing self-powered sensors, working with the use of energy collected from the environment [20–22]: heat, solar, kinetic, etc. Energy self-sufficient sensors strongly simplify the installation and the maintenance procedures of the overall wireless sensor network, as there is no need to access the power grid.

Wireless sensors may be composed of various blocks that depend on the realized functions, required parameters, etc. However, several typical blocks commonly used in such devices may be distinguished: (i) a pollution sensor itself—electronic, chemical, physical, etc.—used to collect data from the external world, (ii) a low pass filter used to prevent the aliasing effect, (iii) an ADC, and (iv) an RF communication block. In state-of-the-art pollution sensors, an additional intermediate analogue-to-analogue conversion may be applied.

An example air pollution sensor was proposed by Texas Instruments in [23]. In this case, an optoelectronic system was directly involved in the detection and measurement of the $PM_{2.5}$ and the $PM_{10}$ particles. The light stream registered by the photodiode was scattered on these particles. The current at the photodiode output was correlated with the concentration of pollutants suspended

in the air. This sensor contained an additional current-to-voltage converter, as the ADC used in this case operates in the voltage mode. We, in turn, propose a current-mode ADC that could simplify this signal processing chain.

In [24], a sensor of $PM_{2.5}$ and $PM_{10}$ particles was reported, in which the zinc oxide-based solidly mounted resonator (SMD) is used. The overall system mounted on a printed circuit board (PCB) also includes an ASIC responsible for the processing of data registered by the $PM_{2.5}$ and $PM_{10}$ sensor. The reported performance shows that the development of specialized chips supporting such systems is moving in the right direction.

Another example is the fully integrated capacitive sensor reported in [25,26]. An important feature of this device, realized in the complementary metal oxide semiconductor (CMOS) 0.35 μm technology, is that the sensor is integrated into a single chip with other data processing components. This approach allows for parallel data processing directly at the output of the sensing block.

In the pollution sensor reported in [27], the level of the $PM_{2.5}$ pollutants is converted into frequency. The output block here is the RS flip-flop that generates a digital 1-bit stream, with a frequency proportional to the pollutant concentration level. For this reason, a conventional ADC is not required here. However, the device is equipped with an additional calibration 5-bit DAC.

Over the past few years, a new trend called edge computing has been observed, where an advanced signal processing is partially performed directly in a sensor, before the results are transmitted to a base station (BS) [28–30]. Those sensors are based on an internal signal data analysis and can make decisions independently from the central station. This feature is useful from the energy consumption point of view, because the RF data transmission block may consume up to 90% of the total energy [31–33]. Currently, low power and low-cost WiFi modules are available; this means that data transmission may be carried out via WiFi networks in urban areas [15,34,35]. The algorithm proposed in the study has a role in analyzing the measured signals already at the sensor level. We focused in particular on two aspects: easy implementation at the transistor level, and resistance on changing noise values in the input signal.

Another important element in the development of air pollution sensors is the performance (accuracy) offered by the sensors used. Accuracy means the consistency of measured pollution levels with their real values [36–38].

The air pollution sensors presented in the literature can be compared according to their various characteristics: their structure, their price, and their measurement accuracy—the collection efficiency (CE). The work [36] presents an overview of various solutions in terms of the measurement accuracy. Particle sensors for $PM_{10}$ offer CE accuracy ranging from 47% to 93%. For particle sensors for $PM_{2.5}$, the CE is in the range between 82% and 93%. The comparison also shows the results for particle sensors for $O_3$, $SO_2$, $NO_2$, CO, and NO. The highest CE was obtained in one of the CO sensors, while the lowest was in one of the $NO_2$ and $O_3$ sensors. They were 99% and 37% respectively. Different CE values may apply to different sensors manufactured by the same company.

In the work [38], comparisons of different sensors are available—their price is in the range of tens to several thousands of euros. However, a high price of sensors does not necessarily indicate their accuracy. The price itself depends on other than the measurement accuracy aspects, for example on their purpose and where they are to be used (e.g., durability in difficult conditions), type of sensor construction, size, etc.

As presented above, microsensors may show various inaccuracies in the measurements of particle levels. Depending on the class of the sensing components, noise may have different levels in comparison with the real signal. We used the reported data to develop a signal that, in turn, is used in tests of the proposed algorithm. In the following sections we examine the behaviour of the designed algorithm for different noise results, even with amplitudes exceeding 70% of the actual signal value. However, such assumptions can be treated as the worst-case scenario. The proposed algorithm enables its calibration at the hardware level. It can adapt to different noise results.

## 3. Materials and Methods

### 3.1. Octave/Matlab Software Development Environment

The proposed algorithm was initially implemented in the Octave environment (the equivalent of Matlab) as part of a larger model that also helps to simulate the aspects that are important from the point of view of hardware implementation: signal resolutions, filter lengths, the values of filter coefficients, etc.

To verify the algorithm, we used real data obtained from air pollution sensors installed in Krakow, Poland. We artificially added noise to these signals, to verify the robustness of the algorithm in the worst-case scenarios. The noise with normal and uniform distributions was applied with the amplitude reaching up to 70%. The algorithm was tested for different configurations that include different types of used filters, their orders and frequency responses, etc.

### 3.2. Chip Design and Verification

We designed a prototype chip containing several crucial components of the sensor in the CMOS 130 nm technology. The system was implemented and verified by the use of selected modules of the Cadence environment: Virtuoso, layout-vs.-schematics (LVS), and Spectre transistor-level simulators. We applied a thorough corner analysis, in which simulations were carried out for the supply voltage and the external temperature varying in between 0.8 and 1.2 V and –40 to +140 °C degrees respectively. Tests were also performed for several transistor models (typical, slow, and fast).

The chip was tested by the use of a dedicated measurement PCB equipped with, among others, an on-board programmable device XC9572XL (Xilinx), a cross-over matrix in the form of a CPLD (complex programmable logic device) circuit, I/V and V/I converters, and buffers for analogue lines.

The PCB cooperated with the measurement equipment, which included: a MyRio 1950 measuring card that operates under the control of the LabView (LV) environment, a programmable, precise DC power supply, and Tektronix oscilloscopes and generators.

### 3.3. Data Used in Testing the Proposed Algorithm

At the current stage of our investigations, a sensor that is able to provide data with sufficient frequency is not available yet. In the present work, we focus on the development and optimization of particular components of such a device. Before the sensor will be finally built as a specialized chip, an optimization and rigorous tests of its particular components have to be carried out. Therefore, the important issue is the availability of reliable test data.

In testing and calibrating the proposed algorithm, we used the real values of the pollution levels taken from the Airly air pollution sensors website (https://airly.eu) [15]. Airly has provided reliable data for many years in major cities in Poland. The data presented in the study were collected in Krakow, one of the largest cities in Poland (800,000 inhabitants), where the air pollution often exceeds the limit values.

The Airly sensors provide measured data every hour (24 samples per day). To verify the proposed algorithm with higher frequencies (e.g., 60 samples per hour), we added samples with interpolated values between the real signal samples. For this purpose, we used a spline function, passing through subsequent samples of the real measurements and then resampled it accordingly with higher frequencies. Then we have added artificially generated noise to the signal, which is equivalent to adding higher frequency components to it. The noise with different amplitudes and distributions (normal or uniform) was applied. During the noise amplitude selection, we based on the results reported in [16], as well as on data reported in [36]. We have selected even higher noise levels than in those two works, which are not expected in the real environment. This allowed for checking the robustness of the algorithm for environmental conditions even more demanding than the real ones.

The algorithm has been tested based on the data from many sensors in a given area. Verification was carried out for several hundred daily recordings from various regions of the city.

## 4. Proposed Contribution to Air Polution Sensor Development

In this section, we present details of the proposed contribution. In the following two subsections, we briefly introduce the concept of high spatial resolution air pollution maps, as well as selected hardware components that may be used in the sensors. In the third subsection, we discuss the proposed algorithm, which may be used to control the communication scheme between the sensor and the external network.

### 4.1. An Air Pollution Monitoring System and High-Spatial-Density Maps with Redundancies

One of the issues is how to increase the resolution of the air pollution maps as well as the frequency with which measured data is provided to a base station. Monitoring of the air pollution is especially important for pedestrians and bicycle traffic participants who, using such maps, can plan both their routes and the time of their journey through the city.

The problem of high levels of pollution applies particularly to areas with dense urban development, usually in the central parts of cities, experiencing increased levels of traffic. Monitoring should be conducted even in particular streets, and measurements should be taken at a relatively low height above the road/pavement. Such conditions pose a serious challenge for solutions where the sensors are powered by energy sourced from the environment (e.g., solar).

For example, if solar energy is to be used to power measuring devices, the issues may include low level of exposure to sunlight. This exposure to sunlight, additionally, varies in different areas at different times. It depends on the geographical arrangement of streets, their bends, or partitions resulting, e.g., from the type of elevation. This is briefly illustrated in Figure 1, for three selected hours, during which the sun is in the east, south, and west.

We propose a redundancy in the number of sensors mounted at various points of the city. In general, this is feasible if appropriate optimization of sensor components towards low power dissipation and smaller sizes is achieved. Thanks to this, their production cost may be significantly reduced. Additionally, it may simplify their installation in the city infrastructure. In the proposed approach, several identical sensors may be mounted in a single point (e.g., a street). They could be installed in such a way that during the day at least one of them is always maximally exposed to sunlight, as shown schematically in Figure 1. For example, sensors No. 2, 3, 5, and 9 are the most active before noon, sensors No. 1, 3 (still), and seven, around noon, while sensors No. 1 (still), 4, 6, and 8, in the afternoon. The air pollution monitoring system configured in such a way is well adjusted to the conditions occurring in densely urbanized areas, particularly those exposed to smog deposition. Figure 1 presents an example urban development, typically dense in central parts of the city. We used data taken from the Google Maps service, that does not distinguish day times.

One of the issues that requires optimization is the number of sensors used in a city. For average-sized cities with a population between 500,000 and a million, this number can range from several hundred to several thousand. In Krakow, Poland, for example, with c. 2000 streets, assuming the described redundancy for each of them, theoretically, 6000 sensors may be required. In practice, however, the described redundancy is not needed for the whole city. It is only required in places with high and dense urban development, i.e., in a relatively small area of the town. The installation of sensors ought to be considered in those areas where pedestrian and cyclist traffic is particularly heavy. Only a limited number of streets are transit streets. For example, in dead-end streets with low traffic, affecting only local residents, there is no need for a high-resolution pollution map.

We assume that the sensors will be energy self-sufficient (no power lines). This feature will facilitate their installation in the urban infrastructure. Similarly, to optimize the system, based on the evaluation of their performance, it will be possible to uninstall and move them to another location easily.

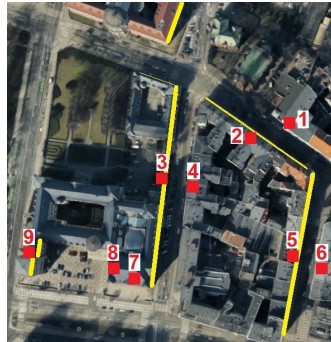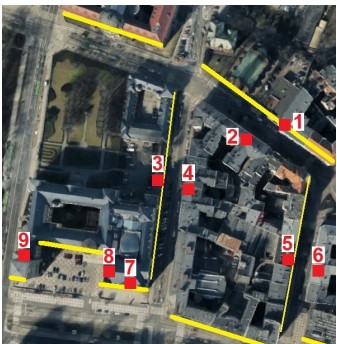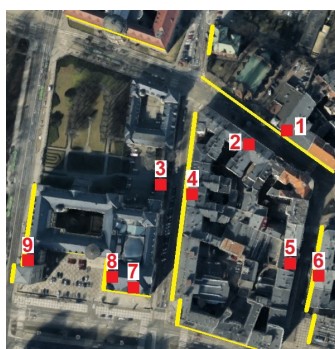

**Figure 1.** Illustration of the availability of solar energy in dense urban areas for several periods of the day (in the morning, at noon, and in the afternoon). The thickness of the yellow lines illustrates the energy level at a given point, while red squares mark the selected positions of the air pollution sensors (prepared on the basis of Google maps street view).

The sensors that offer energy self-sufficiency, including those that work based on light energy, already exist on the market, [19], at the current stage of our investigations we do not perform our own self-sufficiency tests. Currently, we focus on the implementation of the selected components of such a sensor that could also be used in existing sensors of this type.

A particular problem affects the system's operations during the night, especially in the winter, when no solar energy is available. As walking and cycling volumes are low at night, measurements can be taken less frequently, and the sensor may work using the energy accumulated during the day. In such conditions, the algorithm described below could also be deactivated, and the measurement results transferred directly, without their exact processing. This would also reduce energy consumption.

Our goal is not to develop a completely independent system. Alternative networks of such sensors are already operating in many cities. The sensors we are working on may complement existing systems by increasing the resolution of pollution maps in cities mostly in the critical hours of the day when the traffic is large.

### 4.2. Circuit Development

The development of high-resolution maps involves a relatively large number of sensors, which, in turn, requires the development of devices that will be substantially cheaper than, for example, the Airly products today. Such devices will be inexpensive to install and service. These goals may be achieved through the optimization of particular circuit components. For this reason, our investigations focus on ultra-low power and low silicon area solutions.

Reduced size and power dissipation allows for the implementation of a whole block responsible for data processing in the form of a single system-on-chip (SoC). This translates into a reduction in device costs, especially in the large-scale mass production of such sensors. Additionally, it increases the possibilities of their operation using energy sourced from the environment. And this fact leads to a significant reduction in the costs of the installation of sensors in urban infrastructure, with no need to provide a power line [39].

Given the above, we assume a mode of operation where the sensor measures the level of air pollution once per minute. In the meantime, the device is in standby mode, consuming only a minimal amount of energy. An additional assumption is that if the level of pollution does not change in any significant way, the sensor does not need to communicate with the external wireless network. The application of an appropriate algorithm, following the results of data analysis, will control the frequency of communication sessions with the external system.

We present such an algorithm in this paper. Its inputs are samples of the measured pollution stored in the device's internal memory. After each subsequent measurement, the updated data set is subject to analysis. It requires several operations. One of the basic ones is the filtering performed in

custom-developed digital filters—finite impulse response (FIR) and infinite impulse response (IIR). We use low pass and high pass filters of a low order to facilitate their hardware implementation. The filtering is necessary to remove the noise from the recorded data series. At further stages, the computations of fundamental statistical quantities (the mean and the variance) are performed. As a result, we obtain several additional waveforms. Based on these, the sensor assesses whether the level of pollution tends to increase, decrease, or remain at a fixed level.

### 4.2.1. Filtering

In the proposed algorithm for smoothing and averaging operations, we use low-pass FIR filters with a flat frequency response or with equal coefficients of orders, $N$, in-between 1 and 15. The advantage of such filters is their low complex coefficients implemented in the hardware. Owing to this fact, the whole filter can be applied, using only a few asynchronous summing circuits and shifting the bits (equivalent to division operation). In the proposed implementation, we use filters working in parallel and asynchronously. It further simplifies the structure of these sensor components.

We used programmable filters in the algorithm, shown in Figure 2, that can be easily switched to suit different values of the filter order. Each of the visible intermediate blocks in this circuit is a $1^{st}$ order filter with equal coefficients $[1, 1]$. Only one of the configuration bits $m_d$ may equal '1' in any moment. These parameters turn on or off alternative signal paths, thus directing the outputs from particular intermediate $1^{st}$ order sections to the output of the whole circuit. For example, for $m_2 = 1$, the output of the $2^{nd}$ section is provided to the circuit output. The resultant transfer function results, in this case, from the convolution of transfer functions of two $1^{st}$ order sections—the result is $[1, 2, 1]$. In general, to obtain a $D^{th}$ order filter, the $m_D$ signal is set to '1'. The output signal needs to be additionally shifted by $D$ positions to the right. This corresponds to a division by the $2^D$ factor.

### 4.2.2. Hardware-Efficient Computation of Selected Statistical Quantities

Since the measurement samples appear sequentially in the sensor, we applied iterative methods to calculate the statistical quantities used in the algorithm. We introduced some modifications to the existing methods that are suitable for transistor-level realization. We observed that these modifications introduce some error that is insignificant in the described algorithm [40].

Conventional iterative algorithms for the computation of the mean and the variance are presented below [41]. The values of these two quantities are initially set as follows:

$$\text{Mean}_1 = x_1 \tag{1}$$

$$\text{Var}_1 = 0. \tag{2}$$

In subsequent iterations, $n$, these quantities are computed, respectively, as follows:

$$\text{Mean}_n = [\text{Mean}_{n-1} \cdot (n-1) + x_n]/n \tag{3}$$

$$\text{Var}_n = [\text{Var}_{n-1} \cdot (n-1) + (x_n - \text{Mean}_n)^2]/n. \tag{4}$$

Regard the hardware implementation, the problem is the division operation by the number of samples, $n$. For this reason, we introduce the following modifications:

$$\text{Mean}_n = \text{Mean}_{n-1} + [x_n - \text{Mean}_{i-n}]/K \tag{5}$$

$$\text{Var}_n = \text{Var}_{n-1} + [(x_n - \text{Mean}_{i-n})^2 - \text{Var}_{i-n}]/K. \tag{6}$$

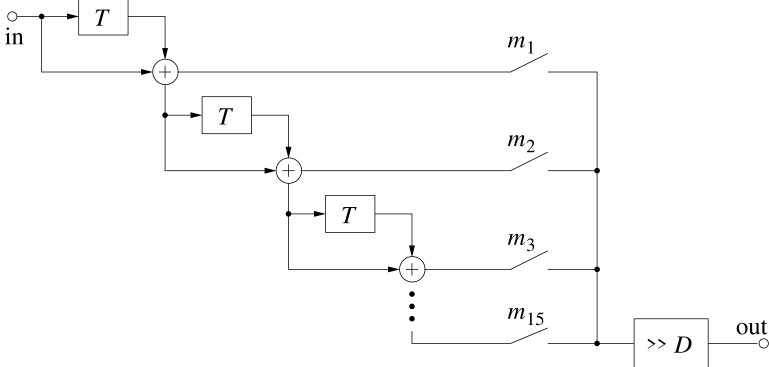

**Figure 2.** Structure of the programmable filter with flat frequency response.

The $K$ variable is set to a constant value, equal to one of the powers of 2. As a result, the division operation is accomplished by shifting the bits in the numerator by $k$ positions to the right, where $k = \log_2 K$. Assuming a constant value of $K$, the mean and the variance are computed in a time window, with the number of samples equal to $K$. The formulas obtained in this way are a rough estimate of the mean and the variance. However, the discrepancy between this particular approach and the conventional one is acceptable here. Due to using a window, these quantities change over time, which is a desirable effect in the proposed algorithm.

The numerators in the formulas above may be either positive or negative. They are coded using the two's complement code, in which the most significant bit (MSB) is '1' or '0', for negative and positive values respectively. For this reason, during the division operation (shifting the bits), the $k$ most significant bits become equal to either '1' or '0', depending on the sign.

### 4.2.3. Current-Mode, Low Chip Area ADC

The state-of-the-art study on air pollution sensors shows that ADCs belong to basic blocks used in these devices. For this reason, in our work, we also focus on the development of ADCs. We deal with converters working in the current mode [42]. A prototype ADC of this type, developed by us, is based on the successive approximation register (SAR) architecture. In the current-mode approach, the DAC is realized as a binary-weighted multi-output current mirror. It is one of the main components of the converter. It provides a reference current that is then compared with an analogue input signal.

In this work, we present a sensor component only briefly, as its structure has been discussed in one of our previous articles, [42]. Here we present the selected, recently obtained measurement results. The proposed current-mode converter may be applied, for example, in a signal processing scheme similar to the one described in [23]. In this solution, a direct analogue output signal of the sensing component is current, which is inversely proportional to the pollution level. The use of the current-mode ADC may simplify the overall signal processing scheme, as the intermediate current-to-voltage converter may be eliminated in this way.

The ADC is still under development and its optimization continues. In Figure 3 it is shown together with an internal multi-phase clock generator that is used to control the course of the SAR algorithm, as well as the proposed algorithm presented below. The ADC offers a low chip area below 0.01 mm$^2$ for 10 bits of the resolution. However, proper measurement results—linearity of the DAC—have been achieved for 7 bits. Generally speaking, it is sufficient; however, we are currently working on the next prototype of this circuit, equipped with a correction mechanism to increase the real resolution.

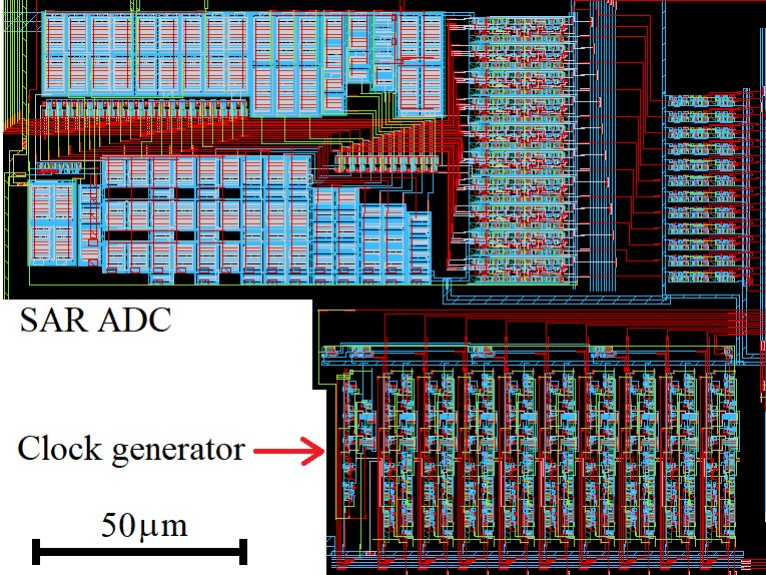

**Figure 3.** Current-mode successive approximation register analog-to-digital converter (SAR ADC) developed for application in low power sensors with an internal clock generator used to control the conversion process.

### 4.3. On-Chip Data Analysis Algorithm for Air Pollution Sensors

The primary solution presented in this work is a prototype algorithm whose role is to control the block responsible for wireless data transmission of collected measurement data. The algorithm makes a decision based on the analysis of the sequence of measurements of the pollution levels. The role of the algorithm is the detection of time in which the level of pollution does not change or changes slightly. During these periods the communication with the base station is either turned off or is less frequent.

When designing the algorithm, several important assumptions were made. One of them was to assure appropriate flexibility, i.e., the ability to modify its internal configuration, depending on the parameters of the processed signal (noise levels and rate of signal changes). It mainly concerned the orders of the used filters and the width of the window used in the computation of the statistical quantities.

The second assumption was to limit the group delay introduced by these components as much as possible. The delay that may be expressed in the number of the samples depends on the order of particular filters. The use of the sensors in the system is almost in real time.

Another assumption was to achieve a low circuit complexity. A vast majority of the blocks used in the proposed algorithm had been designed as parallel and asynchronous circuits, which translates into a simplified internal control scheme. Due to relatively low input data rates (one measurement per minute), focusing on achieving high circuit speeds is not critical here. In this situation, it is possible to reduce the supply voltage of the digital part of the sensor. The power dissipation, $P$, decreases when lowering the supply voltage. Despite the fact that the calculation time, $t$, increases in this case, its increase is slower than the drop of the dissipated power. As a result, the consumed energy $E = P \cdot t$ still decreases [43].

The algorithm is based, to a high degree, on statistical quantities such as the mean and the variance. The proposed methods for the computation of these quantities, discussed above, can be easily implemented at the transistor level, as the complex division operation is eliminated.

A general diagram of the algorithm is shown in Figure 4. Most of the computations are carried out in two parallel channels, although some operations are common for both of these paths. The input signal, $y_{\text{Meas}}$, is provided to a low-pass FIR filter (LPF1) of low order ($N_{\text{lpf1}}$) with the coefficients {1, 1} or {1, 2, 1}. Their values are divided by the sum of the coefficients.

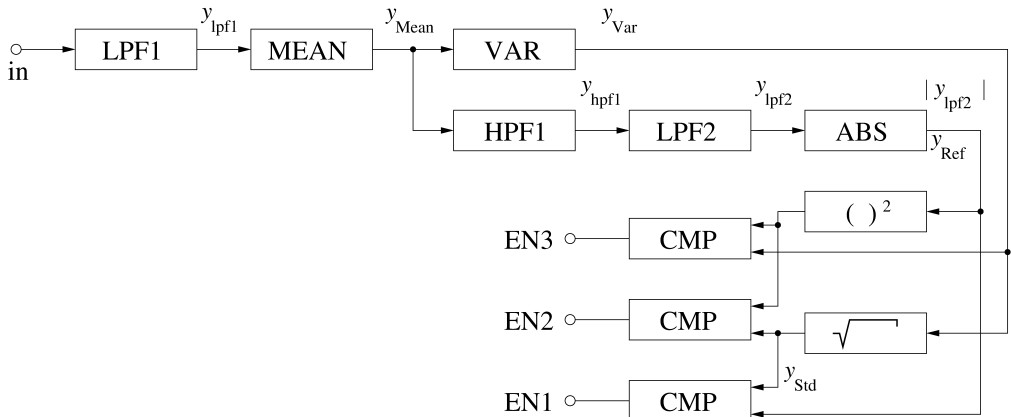

**Figure 4.** Structure of the proposed algorithm supporting the radio-frequency (RF) communication system.

The LPF1 filter computes a signal, $y_{lpf1}$. If the noise level is not high, this step may be omitted. In such a situation $y_{lpf1} = y_{Meas}$. The $y_{lpf1}$ signal is provided to the block that returns the mean value, $y_{Mean}$, in accordance with the formula (5). The MEAN block is, in practice, a low pass IIR filter, in which a new sample of the $y_{Mean}$ signal is computed based on its previous value and a new sample of the $y_{lpf1}$ signal.

After the MEAN block, both calculation paths separate. In one of them, the Mean signal is provided to the VAR (variance) block that computes the variance, $y_{Var}$, in an iterative fashion using the equation, (6). One of the options is to calculate the standard deviation; additionally, however, in this case, an additional rooting block (SQRT) is needed. At the output of the SQRT block, we receive the $y_{Std}$ signal. The $y_{Var}$ and the $y_{Std}$ signals are then used as thresholds at the following steps of the algorithm. These signals are compared with a reference signal (or its square) computed in the second, parallel, computation channel.

In the second path, the output of the MEAN block is filtered with the use of a high-pass (HP) FIR filter with the coefficients {0.5, –0.5} ($N_{hpf1} = 1$), returning a derivative signal $y_{hpf1}$. This signal is then averaged by a subsequent low pass filter (LPF2) of order $N_{lpf2}$, providing a signal, $y_{lpf2}$, at its output. The $N_{lpf2}$ is one of the parameters of the algorithm; however, its value should be selected to be close to the value of the denominator $K$ in (5). Then the magnitude of the $y_{lpf2}$ signal is computed in the absolute (ABS) operation (ABS block). The output of the ABS circuit becomes the reference signal, $y_{Ref}$, which is compared with one of the threshold signals, ($y_{Var}$ or $y_{Std}$). Finally, the output of the comparator may be directly used as the enable (EN) signal by the block responsible for managing the data transmission. For EN = 1 the transmission is allowed.

The described comparison (CMP) operation needs an explanation. For the test purposes, we perform several operations of this type (we call them approaches later in the work), using different input signals, as follows:

$$EN1(n) = cmp(K_1 \cdot y_{Ref}(n), y_{Std}(n)) \tag{7}$$

$$EN2(n) = cmp(K_2 \cdot y_{Ref}(n)^2, y_{Std}(n)) \tag{8}$$

$$EN3(n) = cmp(K_3 \cdot y_{Ref}(n)^2, y_{Var}(n)) \tag{9}$$

where $y_{Std}$ is the square root of the $y_{Var}$ signal. Theoretically the operation given by (7) is an equivalent to the operation of (9). In practice, we use an approximate of the square root operation, so that the results may differ to some extent [44].

Several parameters were used in the described solution: Values of the denominator in Equations (5) and (6), denoted as $K$, and orders $N$ of particular filters. The approach is equivalent to the values of the

Mean, and the Var variables were calculated in a window with a length of *K*. The MEAN and the AVR blocks used in the first channel of the algorithm behave as filters, thus introducing a delay. For this reason, the orders of filters in the second channel should be selected so as not to cause a discrepancy in the time delays between both channels. Taking the specificity of the input signal into account, we usually set these parameters to either 4, 8, or 16, depending on the noise level.

We denote it as a safety mechanism. Its structure is based on a digital counter and a digital comparator. The counter counts cycles (one cycle corresponds to one input sample), for which a given example of the EN signal equals '0'. Once the EN signal becomes '1', the counter is reset. Additionally, when the counter state reaches an assumed threshold, the built-in safety mechanism forces the EN signal to be set to '1' for one cycle. This prevents a situation in which the sensor does not communicate with the external network for a long time.

## 5. Results

Below we present the results obtained during the investigations carried out with the prototype solutions described above. First, we briefly show selected measurement results of the designed current-mode ADC. We focus on input–output characteristics of the DAC, as this component has a direct impact on the performance of the ADC. In the following part, we present the test results of the proposed algorithm.

### 5.1. Laboratory Tests of the Current Mode DAC and ADC

The converter was measured for different values of the input signal range and for different output resolutions. This is possible as the SAR algorithm may be stopped at any conversion step, depending on what is needed. The structure of the implemented ADC allows us to control the upper limit of the input current in a wide range—between 2 and 20 µA. Such a flexibility is achievable only in the current-mode approach.

Figure 5 presents selected measurement results for the input current varying in the range from 0 to 2.5 µA. To facilitate the analysis of the results, the input–output characteristic was normalized in such a way as to achieve a unity slope of the input–output characteristic of the DAC.

### 5.2. Verification of the Proposed on-Chip Algorithm

Figures 6–8 present selected investigation results for $N_{lpf2}$ = 8 and the noise levels doubled in each following case. Diagram (a) presents the original input signal with the noise (dot waveform), as well as the signal computed by the statistical MEAN block. This diagram provides also three EN signals computed according to Equations (7), (8) and (9) (v1, v2, and v3), to facilitate the analysis. Diagrams (b), (d), and (f) show the signals that are directly used to compute the EN signals in particular cases. Diagrams (c), (f), and (g) present the resultant signal at the output of the sensor.

For a better illustration, each of the described diagrams also presents its corresponding EN signal, as well as the output of the safety mechanism. If a given EN signal equals '1', the output of the sensor follows the output signal of the MEAN block. If a given EN signal becomes '0', the internal counter starts counting the inactivity cycles. After a given number of samples (20 in the presented cases), the safety mechanism activates the RF communication block. The measured and processed data is sent to the base station, while the counter is reset.

Figures 9 and 10 present selected results, corresponding to those presented in the previous three Figures, for $N_{lpf2}$ equal to 4 and 16, for the same levels of the noise.

The EN signals are not shown to scale. Their value can be '1' or '0'. Therefore, in diagrams they are multiplied by a constant only to facilitate the presentation.

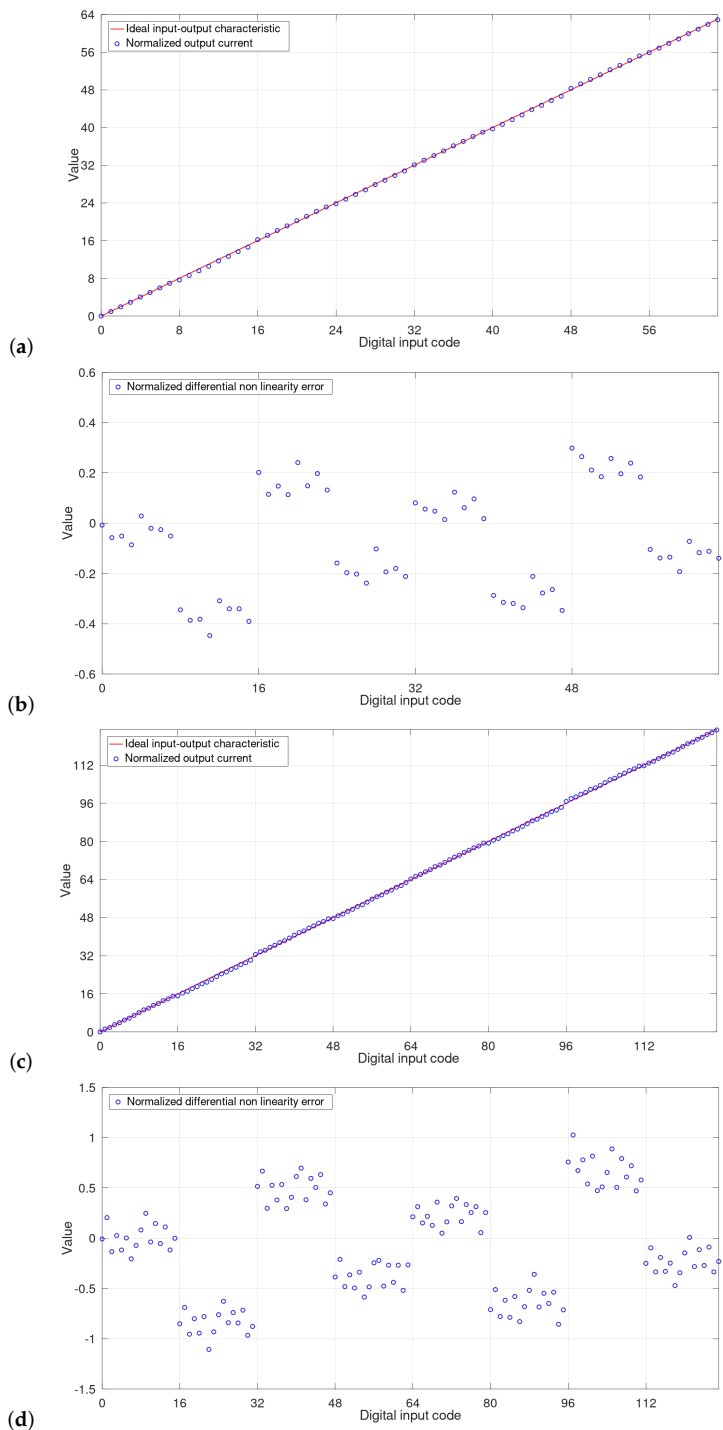

**Figure 5.** Selected measurement results of the prototype current-mode digital-to-analog converter (DAC) implemented in the CMOS 130 nm technology: (**a**) Input-output characteristic for the resolution of 6 bits; (**b**) Normalized differential non linearity error for the resolution of 6 bits; (**c**) Input-output characteristic for the resolution of 7 bits; (**d**) Normalized differential non linearity error for the resolution of 7 bits.

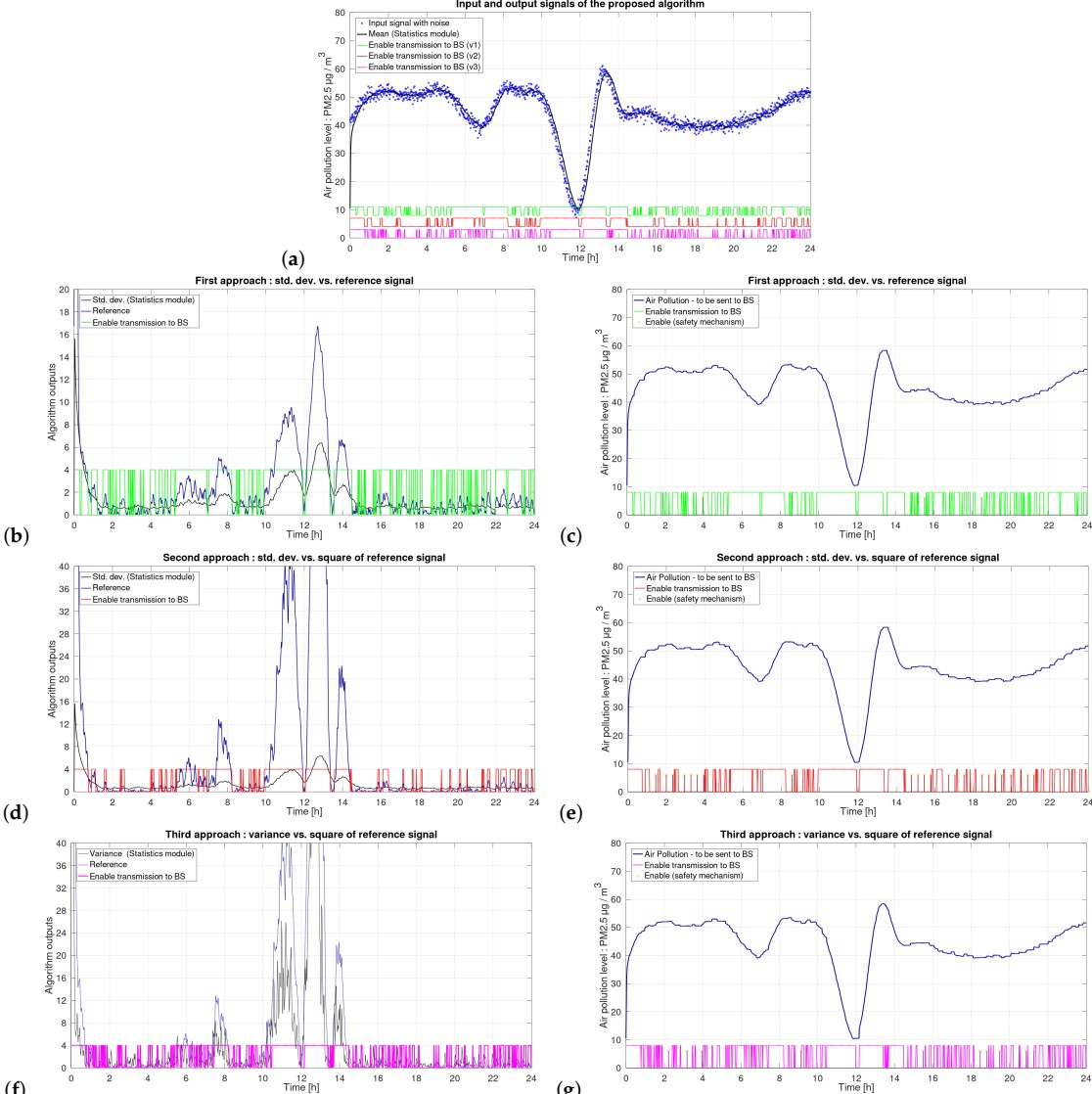

**Figure 6.** Simulation results of the proposed algorithm for small noise levels of c. 6 PM2.5 $\mu g/m^3$ (10%) and the values of $K$ and $N_{lpf2}$ of 8. (**a**) Input signal with noise, corresponding mean signal, as well as transmission enable EN1, EN2 and EN3 signals, for three approaches given by Equations (7), (8) and (9), respectively; (**b**) Standard deviation, reference ($y_{Ref}$) and EN1 signals; (**c**) Sensor output signal transmitted to the base station (BS) for approach 1; (**d**) Standard deviation, square of reference ($y_{Ref}^2$) and EN2; (**e**) Sensor output signal transmitted to the BS for approach 2; (**f**) Variance, square of reference ($y_{Ref}^2$) and EN3 signals; (**g**) Sensor output signal transmitted to the BS for approach 3.

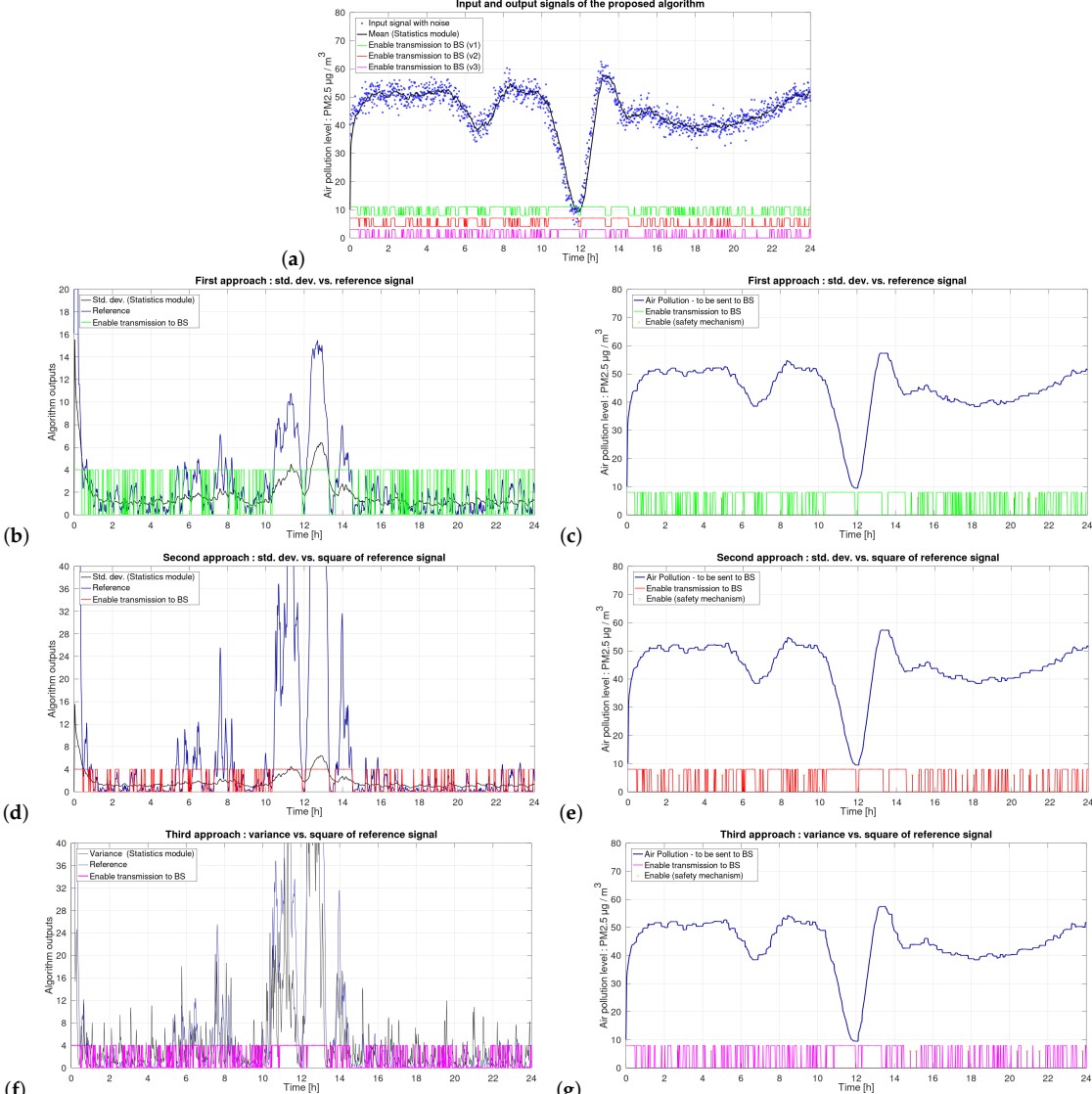

**Figure 7.** Simulation results of the proposed algorithm for medium noise levels of c. 12 PM2.5 μg/m$^3$ (20%) and the values of $K$ and $N_{\text{lpf2}}$ of 8. (**a**) Input signal with noise, corresponding mean signal, as well as transmission enable EN1, EN2 and EN3 signals, for three approaches given by Equations (7), (8) and (9), respectively; (**b**) Standard deviation, reference ($y_{\text{Ref}}$) and EN1 signals; (**c**) Sensor output signal transmitted to the base station (BS) for approach 1; (**d**) Standard deviation, square of reference ($y_{\text{Ref}}^2$) and EN2; (**e**) Sensor output signal transmitted to the BS for approach 2; (**f**) Variance, square of reference ($y_{\text{Ref}}^2$) and EN3 signals; (**g**) Sensor output signal transmitted to the BS for approach 3.

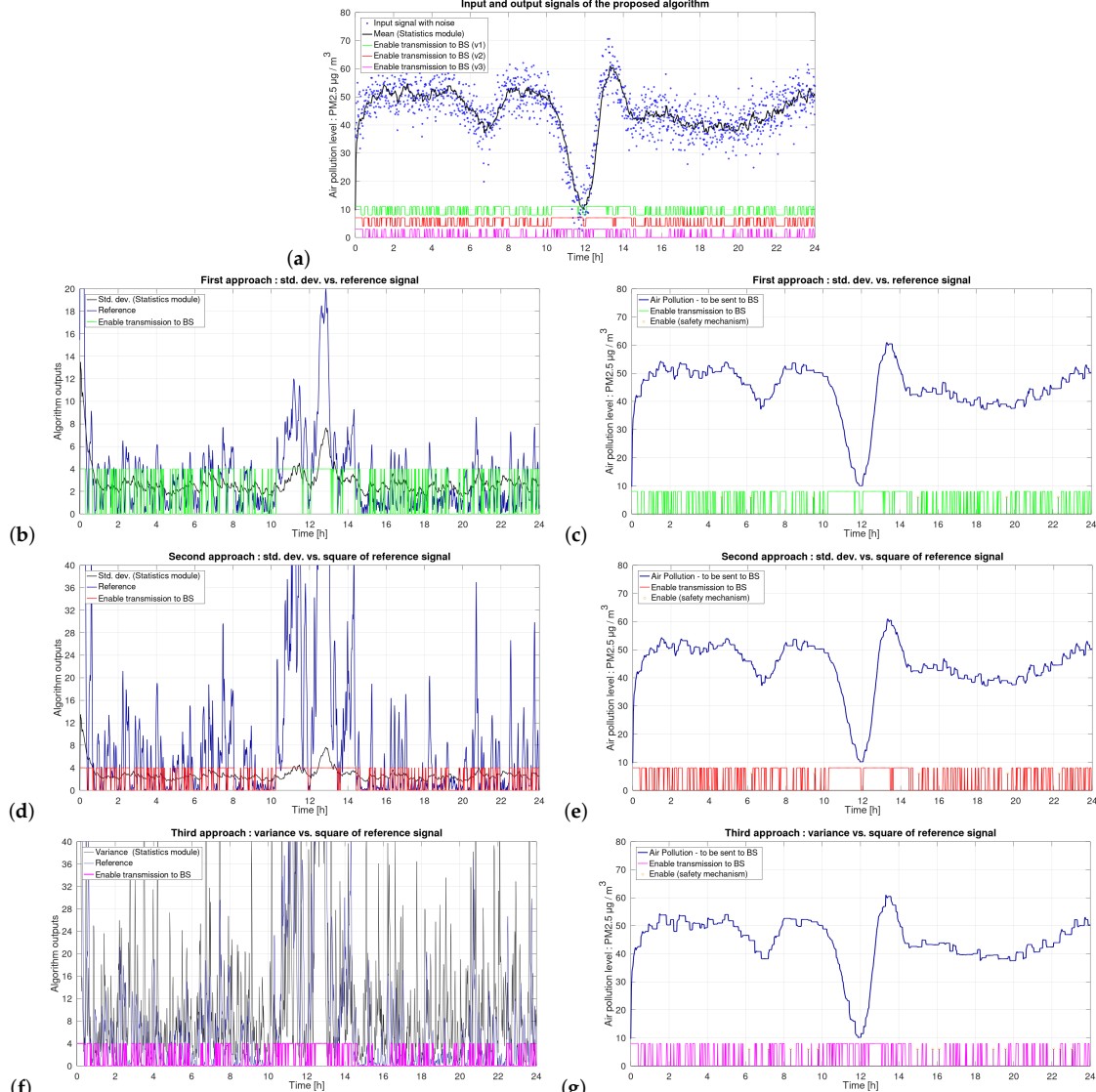

**Figure 8.** Simulation results of the proposed algorithm for large noise levels of c. 24 PM2.5 μg/m$^3$ (40%) and the values of $K$ and $N_{\text{lpf2}}$ of 8. (**a**) Input signal with noise, corresponding mean signal, as well as transmission enable EN1, EN2 and EN3 signals, for three approaches given by Equations (7), (8) and (9), respectively; (**b**) Standard deviation, reference ($y_{\text{Ref}}$) and EN1 signals; (**c**) Sensor output signal transmitted to the base station (BS) for approach 1; (**d**) Standard deviation, square of reference ($y_{\text{Ref}}^2$) and EN2; (**e**) Sensor output signal transmitted to the BS for approach 2; (**f**) Variance, square of reference ($y_{\text{Ref}}^2$) and EN3 signals; (**g**) Sensor output signal transmitted to the BS for approach 3.

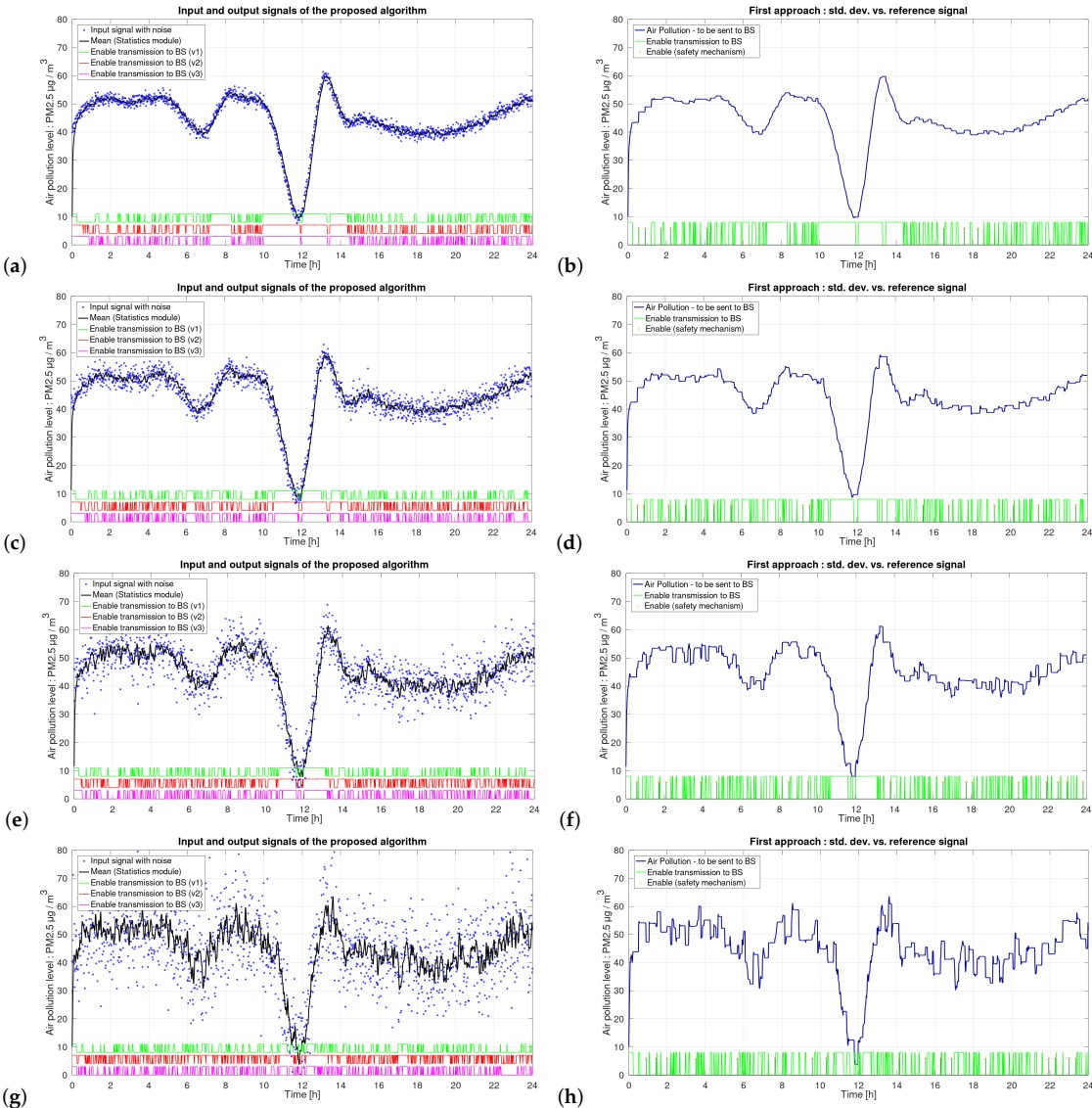

**Figure 9.** Simulation results for Approach 1, for different maximum values of the noise level and the values of $K$ and $N_{lpf2}$ of 4. (**a**) Input, mean, and EN signals for noise amplitude of c. 6 PM2.5 µg/m$^3$ (10 %); (**b**) Sensor output signal sent to BS; (**c**) Input, mean, and EN signals for noise amplitude of c. 12 PM2.5 µg/m$^3$ (20 %); (**d**) Sensor output signal sent to BS; (**e**) Input, mean, and EN signals for noise amplitude of c. 24 PM2.5 µg/m$^3$ (40 %); (**f**) Sensor output signal sent to BS; (**g**) Input, mean, and EN signals for noise amplitude of c. 48 PM2.5 µg/m$^3$ (80 %); (**h**) Sensor output signal sent to BS.

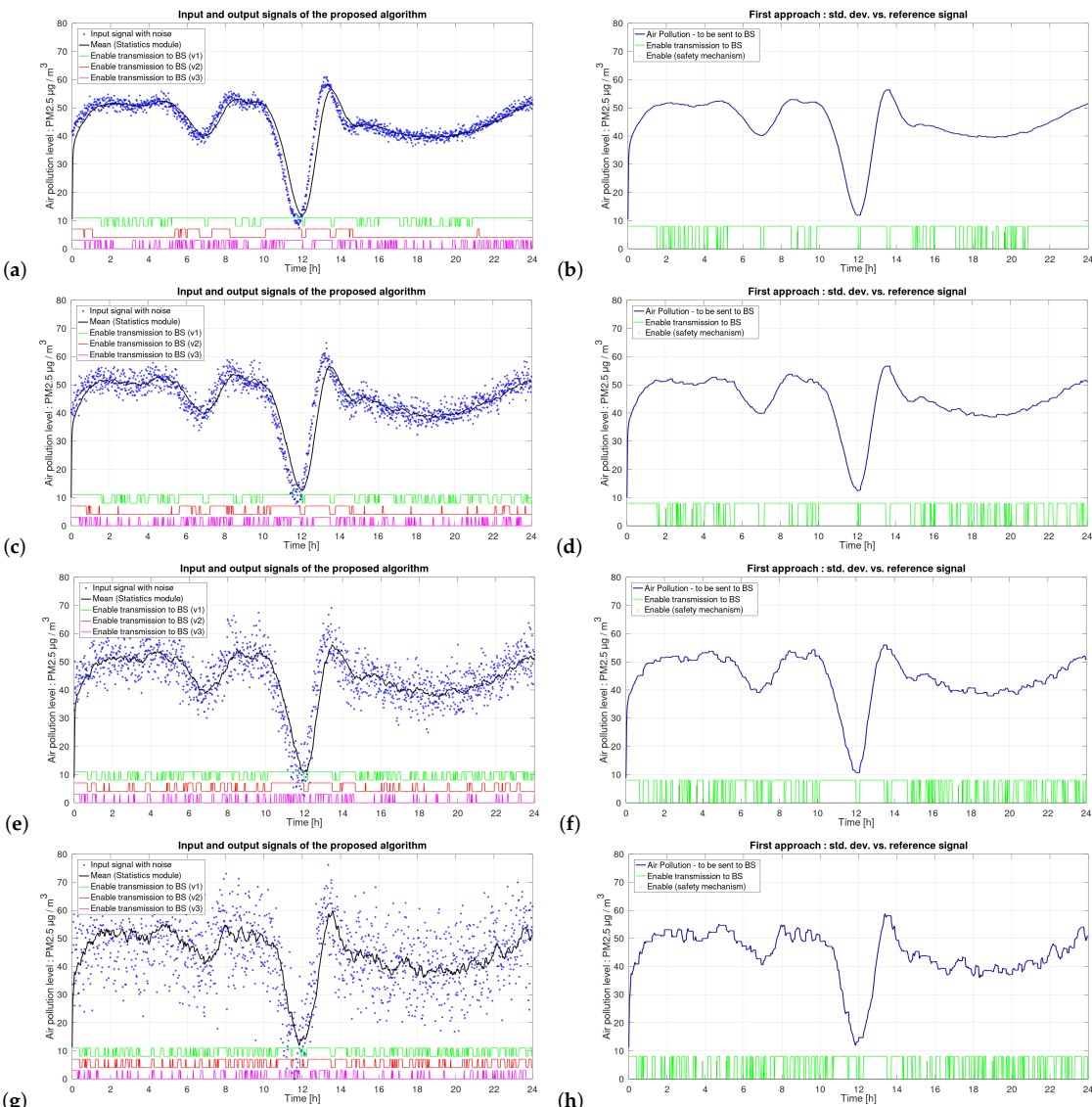

**Figure 10.** Simulation results for Approach 1, for different noise levels and the values of $K$ and $N_{\text{lpf2}}$ of 16. (**a**) Input, mean, and EN signals for noise amplitude of c. 6 PM2.5 µg/m$^3$ (10%); (**b**) Sensor output signal sent to BS; (**c**) Input, mean, and EN signals for noise amplitude of c. 12 PM2.5 µg/m$^3$ (20%); (**d**) Sensor output signal sent to BS; (**e**) Input, mean, and EN signals for noise amplitude of c. 24 PM2.5 µg/m$^3$ (40%); (**f**) Sensor output signal sent to BS; (**g**) Input, mean, and EN signals for noise amplitude of c. 48 PM2.5 µg/m$^3$ (80%); (**h**) Sensor output signal sent to BS.

## 6. Discussion

### 6.1. Current-Mode Digital-to-Analogue Converter

The DAC is one of the main components of the SAR ADC that will be used in the sensor. As the ADC is responsible for providing a digital counterpart of the measured analogue signal that is proportional to the pollution level, it may also impact the quality of the overall sensor device.

The initial requirements for the resolution of the output signal of the ADC were defined at the level of 5 to 7 bits. One of the reasons was that the proposed current-mode solution was an innovative one, and therefore required careful verification. We did not find a direct reference point as the majority of the SAR ADCs reported in the literature operate in the voltage mode. One of the key reasons for

applying the current mode was to achieve a very small chip. In some systems that feature massive parallel processing of analogue signals, it is useful to use multiple ADCs working in parallel.

For research purposes, we finally designed a programmable 10-bit ADC with the chip area of only 0.01 mm$^2$. This value may be compared, for example, with the chip areas of 0.0516 and 0.046 mm$^2$ for the ADCs reported in [45,46], respectively. The converter reported in [45] was designed in a comparable CMOS 130 nm technology, while the one reported in [46] was designed in a more dense 55 nm process.

The reconfigurable structure of the proposed ADC allows the conduction of tests for various signal resolutions, including those initially required. In Figure 5, we present selected measurement results (input–output characteristic of the DAC) for the configurations of 6 and 7 bits. A systematic error for selected digital codes at the level of about 0.5% and 1% can be observed—Figure 5b,d. The source of these nonidealities is an effect of the transistor mismatch, which appeared during the chip fabrication.

The obtained results may be assumed to be sufficient in the light of the precision offered by state-of-the-art air pollution sensors at levels up to 82%–93%, as reported in [36]. One of the advantages of the proposed solution is the almost zero value of the integral nonlinearity (INL) of the DAC—the parameter responsible for general linearity of the sensor. However, the differential nonlinearity (DNL) can be improved if more precise sensing devices are available.

*6.2. The Proposed on-Chip Signal Processing Algorithm*

The work presented in [16] is important, as it demonstrates a series of recorded air pollution levels for measurements carried out in a minute cycle. The authors of [16] additionally compared their results with a reference sensor, in which noise is visible. One of the possible reasons for the visible scatter of the signal samples may be that the measurements were taken in various areas of the city as the bus moved. We expect smaller noise amplitudes with sensors installed in fixed locations. However, taking the reported values, we assumed noise amplitudes between 5% and 80% of a maximum value, including a safety margin. This is visible in Figures 6–10.

As described earlier, the role of the proposed algorithm is twofold. On the one hand, it consists of noise cancellation. On the other hand, based on the smoothed output signal, the algorithm decides when the recorded and processed data should be sent to the base station.

The offered noise cancellation is at the level of 23 dB for the noise amplitudes at the level up to 70%. In our opinion, this result is quite good considering other aspects that are equally important—the delays introduced by the applied filters, as well as the hardware complexity. These results were obtained for simple filters, in which the coefficients of the transfer function were small integers and always positive values. This simplifies the realization of the overall system.

As for the second task of the algorithm, different signals were used to calculate particular EN signals, which directly control the RF transmission. To one of the inputs of the comparator, we provided either the variance or its root, the standard deviation. At the second input, it was either directly the $y_{\text{Ref}}$ signal, or its square. For higher noise amplitudes, the obtained results were similar. At lower noise values, the EN1 and the EN3 signals were more sensitive to changes in pollution levels. Both these signals are counterparts. For this reason, they exhibit comparable behavior. Relatively small differences result from the way the square root operation is performed when calculating the EN1 signal. When computing the EN2 signal, the $y_{\text{Dev}}$ signal is compared with the square of the reference signal, $y_{\text{Ref}}$. The square values of particular samples of the $y_{\text{Ref}}$ signal exhibit greater amplitude fluctuations than of the signal without squaring. For the values smaller than 1, the squared values are smaller than the values of a not squared signal. The squared signal is more sensitive to differences in the mean signal.

*6.3. Circuit Complexity and Energy Consumption*

The proposed algorithm (Figure 4) consists of several FIR filters, a block calculating the average value of samples (MEAN), a block calculating the variance (VAR), digital comparators (CMP), and

blocks calculating the square and the root. A basic block used in both low-pass filters (LPF1 and LPF2) is filtered with order $N = 1$ with equal coefficients. It calculates the average value of two adjacent input samples. Let us denote it as $LPF0_n$, where $n$ is the resolution of the signal at its input. This filter consists of a memory block (MEM) containing $n$ memory cells and an asynchronous $n$-bit asynchronous adder (MBFA – multi-bit full adder). The memory cells are built based on D-flip flops, composed of 26 transistors, while the MBFA consists of $n$ 1-bit full adders (1BFA). A single 1BFA is composed of 28 transistors.

The LPF1 and LPF2 filters also use $n$ switches, which are controlled by the $m_i$ signals. A single switch was implemented as a transmission gate consisting of two transistors – NMOS and PMOS (N-type and P-type metal-oxide-semiconductor). In the real implementation, the bit shift operations (equivalent to division) are performed immediately after particular switch blocks. The bits for a given filter stage are shifted always by some value that does not require an additional block.

Taking the numbers above, the $LPF0_n$ filter consists of about $n \cdot 56$ transistors. Both the LPF1 and LPF2 filters are programmable. Due to the assumed flexibility of the algorithm, these filters consist of as many LPF0 blocks as the order $N$ of these filters. The LPF1 filter, therefore, contains two LPF0 blocks: $LPF0_8$ and $LPF0_9$. The number of transistors in the LPF1 filter is, therefore $(8 + 9) \cdot 56 = 952$. The LPF2 filter is composed of 15 LPF0 blocks, with the signal resolution at their inputs increasing by one on each subsequent stage. The sum of the memory cells in the filter is equal to the sum of the 1BFAs and the number of switches and can be calculated as the sum of the arithmetic sequence with step 1. It equals 225 or an 8-bit input signal. The overall LPF2 filter, therefore, consists of 12,600 transistors. The high-pass (HPF1) filter has a similar structure to the LPF0 one, i.e., the number of transistors is not exceeding 500.

The comparators are realized as multi-bit full subtractors (MBFS), which have a computational complexity similar to MBFAs. The MEAN block consists of about 1000 transistors, while the VAR one consists of about 4000 transistors due to the need to calculate the square of the signal (Equation (6)). The total number of transistors in the overall algorithm does not exceed 20,000. This translates to the chip area not exceeding approximately 0.1 mm$^2$, for the implementation in the CMOS 130 nm technology. This value was calculated based on a real prototype chip designed by us in this technology. In new technologies, the chip area can be significantly reduced.

One important aspect when assessing the performance of electronic devices is the dissipated power. However, in devices where the energy self-sufficiency is assumed, the consumed energy is a more important factor. Both these parameters, although important, are not critical. This is due to the relatively low required speed of the system. A single iteration of the algorithm will be performed once every minute or several times more often when a new pollution measurement sample is available. The ADC was successfully tested for sampling frequencies up to 0.55 Msamples/s, while the digital blocks up to 100 Msamples/s. This shows, that for the vast majority of the time these circuits are inactive and may be in power-down mode.

In CMOS technologies, these consume negligible energy in such situations. The energy consumed by particular sensor components can be estimated based on the measurement results of the prototype chip. For the ADC, the energy consumed during the conversion of a single measurement of pollutants does not exceed (on average) of 3.5 pJ per bit of the digital output signal. The overall algorithm consumes a maximum of about 19 pJ of energy to process a single signal sample. The results are given for a standard 1.2 V supply voltage in the 130 nm CMOS technology. During the development of the algorithm, we initially tested it with relatively high resolutions of the input signals. However, since the standards are frequently exceeded even by up to 50%–200%, therefore, we assumed that the algorithm accuracy at the level of 98%–99% is sufficient, so smaller signal resolutions at the level of 8 bits may be assumed. It translates into a lower circuit complexity. However, since the algorithm itself works on digital data, it can be easily scaled up in terms of the signal resolution.

A parameter comparison with other state-of-the art solutions may be carried out at this stage of the project for the designed ADC/DAC. In Table 1, our converter was compared with selected SAR

ADCs designed in 130 nm CMOS technology for similar sampling rates based on parameters: the power dissipation, the chip area, data rate, and the speed of action. To facilitate direct comparison, we used a figure-of-merit (FOM), defined as follows:

$$\text{FOM} = P/(2^n \cdot f_S) \tag{10}$$

where $P$ is the power dissipation, $n$ is the resolution at the output of the converter, and $f_S$ is the maximum sampling frequency. We observed that a smaller FOM is associated with a smaller power required to compute a single output code.

**Table 1.** Comparison of low power and low chip area SAR ADCs.

| Ref. | Mode V/I | Res. bits | $f_S$ (MSmpl./s) | $P$ (mW) | A (mm²) | FOM fJ/conv |
|------|----------|-----------|------------------|----------|---------|-------------|
| [47] | V | 10 | 1 | 0.147 | ND | 437 |
| [48] | V | 8 | 1 | 1.2 | 0.1 | 4687 |
| [49] | V | 8 | 0.1 | 0.0032 | 0.08 | 143 |
| This work | I | 8 | 0.55 | 0.0132 | 0.01 | 93.6 |

Limitations and Future Work

One of the problems, common in any system based on filtering, is a delay introduced by filters. For larger noise levels, it is reasonable to select larger values of the $K$ and the $N_{\text{Lpf2}}$ parameters. As a result, a smoother signal is received at the sensor output. This is done at the price of larger signal delays, which translates into the timeliness of the created pollution map. There is, therefore, a trade-off between the measurement accuracy and the map timeliness. A solution may be considered in the case of larger variations of the input signal. The parameters mentioned above are automatically switched to larger values.

To limit the described trade-off between the noise suppression abilities and filter delays, one can consider the implementation of an additional algorithm responsible for a prediction, in a short time horizon, of the pollution levels based on the measured and processed data sequence. At this stage, we assume that such a prediction may be performed in the base station. In further studies, however, an option is to include an additional hardware component that would perform this task directly at the level of the particular sensors. Such a solution would be justified as a system based on such sensors would be more scalable. In practice, this means that the base station's computational load would then be less dependent on the number of sensors that the base station would have to support.

One of the aspects is the universality of the proposed solutions. In general, in our work, we do not deal directly with sensing components of the sensor. The proposed algorithm is designed in such a way as to enable its application in different scenarios. Depending on what is needed, one can use one of the offered functions or both. For example, it can also be used in sensors installed indoors. It may be assumed that in a confined space the energy supply problem may be solved more easily, as there is usually access to the power network. For this reason, the function of management of the RF transmission block may be of secondary importance in this case.

Additionally, in a confined space, the expected noise levels are smaller. The possible modifications of the algorithm would effect the appropriate calibration. There is a possibility of adjusting the parameters of the filters and the components depending on the noise levels.

## 7. Conclusions

The investigations presented in this paper aim at the development of high spatial density air pollution maps for smart cities. One of the milestones of the project is building a fully functional air pollution sensor and its verification in a real environment. Building such a device is a complex

task that must be preceded by optimization and verification of the particular components. We are currently at this stage of the project. For these reasons, in this paper, we present the results for selected components of the sensor. One of them is the algorithm responsible for processing the collected data. The algorithm allows us to achieve data processing distributed over the wireless sensor network and thus fits into the area called edge computing. For a large number of such sensors in a city, this will substantially reduce the amount of data sent over the network and the computational complexity of the algorithms localized in the base station.

The designed algorithm was tested using data that was artificially superimposed with a high amplitude noise. It offers a simple structure, with the number of transistors not exceeding 20,000. Despite this, it can reduce the noise level by more than 20 dB, which translates into the data quality exceeding 99% in the case of typical noise amplitudes.

The paper also briefly presents selected measurement results of a prototype digital-to-analogue converter operating in the current mode implemented with CMOS 130 nm technology. The current mode simplifies the structure of the device working based on a laser sensor, with the current signal as a sensing blocks output. The presented measurement results are for the signal resolutions of 6 and 7 bits. The optimization of this converter is still ongoing toward increasing its real signal resolution. The proposed algorithm and the converter feature relatively low energy consumption (per a single input sample), at the levels of 3.5 and 19 pJ, respectively.

**Author Contributions:** Conceptualization: M.B., R.D. Methodology: R.D., M.B., J.P., T.T. Software: R.D. Validation: R.D., M.B., J.P., T.T. Formal Analysis: R.D., J.P., M.B., T.T. Data Curation: M.B., R.D., Writing—Original Draft Preparation: R.D., M.B., J.P., T.T. Writing—Review & Editing: R.D., M.B., J.P., T.T. Visualization: M.B., R.D. Supervision: M.B. Project Administration: R.D. Funding Acquisition: M.B. All authors have read and agreed to the published version of the manuscript.

**Funding:** This work was co-financed by Ministry of Science and Higher Education of Poland within the frame of projects (no. 0111/SBAD/0293).

**Conflicts of Interest:** The authors declare no conflict of interest.

## Abbreviations

The following abbreviations are used in this manuscript:

| | |
|---|---|
| ABS | Absolute Operation |
| ADC | Analog-to-Digital Conversion |
| ANN | Artificial Neural Networks |
| ASIC | Application Specific Integrated Circuit |
| BS | Base station |
| CMOS | Complementary Metal Oxide Semiconductor |
| CMP | Comparator |
| CPLD | Complex Programmable Logic Device |
| DAC | Digital-To-Analog Converter |
| EN | Enable |
| FIR | Finite Impulse Response (Filter) |
| FPGA | Field Programmable Gate Array |
| HPF | High-Pass Filter |
| I/O | Input–Output |
| IIR | Infinite Impulse Response (Filter) |
| IoT | Internet Of Things |
| ITS | Intelligent Transportation System |
| LPF | Low-Pass Filter |
| LVS | Layout-Vs-Schematics |
| MUX | Multiplexer |
| NMOS | N-type Metal-Oxide-Semiconductor (transistor) |

| OpAmp | Operational Amplifier |
|---|---|
| PCB | Printed Circuit Board |
| PM | Particulate Matter |
| PMOS | P-type Metal-Oxide-Semiconductor (transistor) |
| RF | Radio-Frequency |
| SAR | Successive Approximation Register (ADC) |
| SMD | Solidly Mounted Resonator |
| SoC | System-On-Chip |
| VAR | Variance |
| WHO | World Health Organization |
| WSN | Wireless Sensor Network |

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
