# Peer review of "Hardware Efficient Solutions for Wireless Air Pollution Sensors Dedicated to Dense Urban Areas"

_remotesensing, doi:10.3390/rs12050776_

Round 1

Reviewer 1 Report

The authors have improved the original manuscript up to a publishable form, which fits the Urban Remote Sensing section. 

Author Response

Many thanks for your positive recommendation.

Reviewer 2 Report

Dear Author,

Adapt introduction subdividing into two chapters

1. Introduction: containing the text up to page 2 and its suitability for this purpose

2. Air pollution monitoring systems (software and hardware): Content of item 1.1, as well as figures, technical data such as range, accuracy and operating range, etc.

In chapter 2.1.1. Octave / Matlab software development environment: Needs more details about the data used and the instruments used and their accuracy.

In this chapter, what concerns me the most and details and presents the "design", especially considering the direction of the product towards Hardware Efficient Solutions.

Therefore, the importance is the best definition or the development and presentation of figures, diagrams and details of the calibration and evaluation settings of the real system.

One example is 3.1. Laboratory tests in the current DAC and ADC mode that do not have the same test layout, sensor calibration and measurement errors.

Request the inclusion of a section addressing the comparative validation of the developed system, preferably in the laboratory and in the field. Suggest the inclusion of a communication validation and its wireless range.

Consult the system as all suggestions to include an energy analysis compartmentalized with the state of the art.

Author Response

The reviewer has presented us with important comments and requests, which we believe have improved the quality of our manuscript significantly. In the following pages, we address each and every comment in detail. Each comment is marked by the reviewer’s original text indexed for convenience of reference, and explains what and how changes were implemented in the new submission. We hope the revised manuscript is much closer to the high standard of Remote Sensing.

R1: Adapt introduction subdividing into two chapters

1. Introduction: containing the text up to page 2 and its suitability for this purpose

2. Air pollution monitoring systems (software and hardware): Content of item 1.1, as well as figures, technical data such as range, accuracy and operating range, etc.

A: In the revised version of the manuscript, we divided this section into two, as noted. Our initial structure with a single part resulted from the fact that we tried to stick to the IMRaD principles, in which everything that concerns introduction (including the state of the art study) is usually in a single section.

R2: In chapter 2.1.1. Octave / Matlab software development environment: Needs more details about the data used and the instruments used and their accuracy.

A: We currently do not have our sensor. We have worked on its individual components, taking into account the requirements that can be defined on the basis of commercial solutions and those that have been described in the literature.

Therefore, at this stage of research, we had to rely on the data available on the web. The Airly system has been providing such data for several years in many Polish cities. The company's devices are bought by local authorities and installed in an increasing number of cities. Hence, we found the data provided by the sensor network reliable. It can be assumed that the measurement accuracy of 1% is a reasonable assumption. Manufacturers often give similar accuracy. In Poland, pollution standards are regularly exceeded by several dozen to several hundred percents. Therefore, this area can be considered representative for testing solutions of this type.

The Airly system (the most common in Poland) provides measurement results at hourly intervals. Therefore, to verify the proposed algorithm, we had to increase their frequency to suit our assumptions. For this purpose, we used the spline function, passing through subsequent samples of real measurements. Then we sampled the waveforms again at a higher frequency (one sample per minute). This operation alone would not matter if it wasn't for the addition of higher frequency signal components. Such components are always contained in the noise. Therefore, adding noise to the signal by us increases the reliability of this data. We are aware that despite this, the data is to some extent, artificial. However, to thoroughly check the operation of the proposed algorithm, we selected very high noise levels, which do not occur.

What is important, the algorithm has been tested based on data from many sensors in a given area and in different periods. Verification was carried out for several hundred data recordings. In the revised version of the article, we have expanded the description of the data used in the tests. We created a separate subsection (p. 3.3).

R3: In this chapter, what concerns me the most and details and presents the "design", especially considering the direction of the product towards Hardware Efficient Solutions.

A: In the revised version of the manuscript, we added a new subsection, in which we analyzed the hardware complexity of the algorithm. Based on the prototype integrated circuit we made, it was possible to count the number of transistors and to assess the target chip area (p.6.3).

R4: Therefore, the importance is the best definition or the development and presentation of figures, diagrams and details of the calibration and evaluation settings of the real system. One example is 3.1. Laboratory tests in the current DAC and ADC mode that do not have the same test layout, sensor calibration and measurement errors.

A: We corrected the layout of the Figures presenting the measurement results of the DAC.

R5: Request the inclusion of a section addressing the comparative validation of the developed system, preferably in the laboratory and in the field. Suggest the inclusion of a communication validation and its wireless range.

A: In our investigations, presented in this paper, we follow a design flow typical for complex chip development. In our former VLSI ASIC projects we always stared with designing particular system components as separate building blocks in prototype chips implemented in several CMOS technologies. The problem is that before any devise with the complexity comparable to discussed sensor may be produced, its particular components should be carefully optimized and tested, to avoid risk of errors. The problem is a very high price of such prototypes.

For this reason, we currently do not have a full sensor of our own production yet. Therefore, unfortunately, we are unable to provide test results for an overall sensor in the real environment. As for the matter of the verification of wireless communication range, in most of sensors available on the market today, including the Airly solutions described in the paper, the WiFi communication is applied to transmit data. For this reason, the WiFi communication may be treated as a standard technology, which is already well tested by others. Since we assumed the same approach in our case, we think that such tests may be performed after the final chip will be developed.

However, we already designed a prototype chip, as described in the paper, which contains a majority of the blocks to be used in such a sensor. It includes, among others, the DAC and ADC, as well as digital components, such as: asynchronous multi-bit adders and subtractors, memory blocks, multi-phase clock generator, FIR filters, etc. All these components have been recently tested, so soon the works on the overall sensor itself will start.

We've added more implementation details in the improved version of the manuscript. First of all, we have added a new section with the analysis mentioned above of the hardware complexity of the designed algorithm. Based on the number of transistors and based on the results of the transistor level simulations and measurements of our previous prototype integrated circuits, we were also able to estimate the power consumed by the algorithm in the event of its full implementation in CMOS technology (p.6.3).

R6: Consult the system as all suggestions to include an energy analysis compartmentalized with the state of the art.

A: In this paper, we, in particular, focused on optimizing the proposed algorithm in terms of its ability to cancel the noise and to distinguish periods in which the levels of pollution change quickly. For this reason, most of the presented investigation results are related to the operation of the algorithm at the signal processing level. The power consumption is a significant parameter, although in this particular case, it is not critical. This is due to the fact that both the algorithm and other sensor components do not have to work at high data rates since the measurements are made relatively rarely.

In the revised version of the work, we have added a comprehensive section 6.3, in which we discuss the hardware complexity of the proposed solutions, also adding the energy consumption analysis. For the converter, we compared its main parameters with selected solutions featuring a similar structure (see Table 1).

Reviewer 3 Report

The manuscript titled “Hardware efficient solutions for wireless air pollution sensors dedicated to dense urban areas” by Banach et al. introduces a sensor designed for urban air pollution monitoring. The sensor with the designed algorithm features low energy consumption and efficient reduction of noise. The authors exemplified the performance of the sensor using an artificial dataset based on hourly measurement data. In general, this topic is interesting considering the existing demand of high resolution air pollution mapping in urban areas. However, there exist several issues in the manuscript, and the authors will need to resolve/justify them in the revision. Considering the topic of this work, it may not fit very well with Remote Sensing’s scope. Probably Sensors or Urban Science could be a better option.

Major concerns:

Lines 182: “reaching up to 50%”, while in line 169, “with amplitudes exceeding 70%”.

Figure 1: the subplot in the middle seems incorrect. Shadow for sites 3 and 4 (and 5-6) should be minimal at noon if they are along a north-south street.

Line 224: “that during the day at least one of them is always maximally exposed to sunlight”. How about during nighttime? There is no test about whether the sensor is energy self-sufficient, especially when the sunlight is insufficient or even absent. In addition, such arrangement will likely miss the detailed concentration fields (along the street) when some of the sensors are down.

Lines 238–239: “it will be possible to change their location relatively quickly”. So the sensor network is a mobile one????

Section 3: The tests used interpolated data with artificially generated errors. Is there any way to test the robustness of the sensor with higher temporal resolution data?

Some minor comments on grammatical issues:

Line 2: “implementing” should be “implementation”

Line 32: “respirator”

Lines 39-40: “effects of … was”

Author Response

The reviewer has presented us with important comments and requests, which we believe have improved the quality of our manuscript significantly. In the following pages, we address each and every comment in detail. Each comment is marked by the reviewer’s original text indexed for convenience of reference, and explains what and how changes were implemented in the new submission. We hope the revised manuscript is much closer to the high standard of Remote Sensing.

Major concerns:

R1: Lines 182: "reaching up to 50%", while in line 169, "with amplitudes exceeding 70%" .

A: The first value was a mistake. During the tests, we were modified the amplitude of the noise up to some maximum value. In the initial series of tests, we performed simulations with the noise of up to 50%, but then we extended the amplitude to even 70%. We have corrected it in the manuscript (line 186).

R2: Figure 1: the subplot in the middle seems incorrect. Shadow for sites 3 and 4 (and 5-6) should be minimal at noon if they are along a north-south street.

A: This Figure has been used only an illustration of the urban dense development. We have taken it from Google maps, where unfortunately there are no options for different day times. This is why we added yellow lines to the subplots. A better option would be using a drone and take series photos; however, we do not have access to such a device. We added such a comment to the manuscript (lines 257-258).

R3: Line 224: "that during the day at least one of them is always maximally exposed to sunlight". How about during nighttime?

A: The proposed system is dedicated to users who are most exposed to pollution. The main addressee of the system is, therefore, pedestrians and cyclists who move in the city, often in heavily polluted areas. At night, the traffic of these two groups of city users is negligible in most cities. At the same time, concentration levels are usually much lower at night. We observed this for many points of the city based on data provided by Airly. For this reason, we believe that these sensors can either be inactive at night or take measurements less often, using the energy previously accumulated in the battery.

Our goal is not to treat the network of such miniature sensors completely independently and as the only system working in a given city. Alternative sensor networks are already in operation, but their distribution is rarer. Not everywhere, such networks are available. For example, in addition to the price of the sensor itself, Airly requires a monthly subscription payment for each device. Therefore, this system is not only expensive to buy but also costly to maintain if the number of sensors would be large. The sensors can be a kind of refinement in the sense of map resolution and increasing the frequency of measurements. In this situation, in our opinion, it is not a big problem if, at night, the measurements were carried out less frequently and from not all measuring points. We have completed the description of this part of the work, adding an appropriate explanation (lines 282-285).

R4: There is no test about whether the sensor is energy self-sufficient, especially when the sunlight is insufficient or even absent. In addition, such arrangement will likely miss the detailed concentration fields (along the street) when some of the sensors are down.

A: Currently, we do not have our own complete sensor. This is the main reason why we did not perform self-sufficiency tests. We are currently working on individual components of such a sensor, taking into account the requirements that can be defined based on commercial solutions and those that have been described in the literature.

It is possible to find commercial sensors that are self-sufficient. These include Cypress Semiconductor solutions, which we refer to in our paper, and which work based on light energy. For this reason, we assumed that such self-sufficiency could be realized without dealing with this issue our self. Building the entire sensor is our goal, but first, we need more detailed research with an already designed algorithm to make its possible tuning before full implementation in the integrated circuit. We've added a brief explanation in the revised manuscript (lines 268-275).

R5: Lines 238–239: "it will be possible to change their location relatively quickly". So the sensor network is a mobile one?

A: In the paper, the mobility means easy of installation and uninstallation. We assume that there will be no need to provide the power lines. Therefore, changing the location of the device can be performed relatively easily. This feature can also be useful from a research point of view. You can periodically place the sensor in different places of the street and look for its optimal location.
In the paper, we modified this sentence to make it less confusing (lines 269-271).

R6: The tests used interpolated data with artificially generated errors. Is there any way to test the robustness of the sensor with higher temporal resolution data?

A: Yes, there are such possibilities. We even conducted such research before for many Airly network sensors located in various places of Krakow city (Poland). Ultimately, we assumed that very high signal resolutions would not be necessary, because for a typical user (pedestrian) the accuracy of 1-2% is sufficient. This assumption also results from the observation of how much norms can be exceeded in cities. In Poland, they are frequently exceeded even by up to 50-200%. In this situation, the accuracy of the measurement error of 1-2% is not a big problem. This translates into a signal resolution of 7 or 6 bits, respectively. Hence the design of our ADC and DAC, in which the signal resolution at this level was achieved.

The algorithm itself works on digital data; it can be easily scaled. The used filters are based on digital multi-bit full adders, the resolution of which can be easily extended by adding subsequent asynchronous one-bit adders.

The assumed resolutions of 7-8 bits are also due to circuit complexity optimization. In our opinion, if a given resolution is sufficient from practical reasons, then there is not recommended to unnecessarily increase the complexity of the circuit. In the revised version of the manuscript we have added a comment on that (p.6.3).

R7: Some minor comments on grammatical issues:

* Line 2: "implementing" should be "implementation"

It has been corrected.

* Line 32: "respirator"

We removed this part of the sentence, as it could be confusing.

* Lines 39-40: "effects of ... was"

It has been corrected.

Reviewer 4 Report

The authors propose the design and implementation of hardware components of low power air pollution sensors. The components are formed by a processing algorithm, analogue-to-digital (ADC) and digital-to-analogue converters (DAC) and filters.
I find the work interesting and useful. Related work is well described and updated.
However, I think that the manuscript should be better organized and written
My recommendation is to reconsider after major revisions because some aspects have to be improved:

- The authors said that the number of wireless communication sessions can be limited based on collected data analysis or that the algorithm controls the wireless data transmission of collected measurement data. However, the reviewer thinks it is not a novelty. A simple algorithm can be used to send data when is necessary or there are value variation. It should be re-written. In fact, it is not discussed in Discussion section.
- At the end of section 1, a paragraph about the structure of the contribution should be described
- Section 1.1 should be numbered as 2. State of the art or Related work
- Section 2.2 is too long. It should be considered as a new section, for example Section 4
- The Conclusions should be rewritten because the reviewer finds them vey generic.

Author Response

The reviewer has presented us with important comments and requests, which we believe have improved the quality of our manuscript significantly. In the following pages, we address each and every comment in detail. Each comment is marked by the reviewer’s original text indexed for convenience of reference, and explains what and how changes were implemented in the new submission. We hope the revised manuscript is much closer to the high standard of Remote Sensing.

R1: The authors said that the number of wireless communication sessions can be limited based on collected data analysis or that the algorithm controls the wireless data transmission of collected measurement data. However, the reviewer thinks it is not a novelty. A simple algorithm can be used to send data when is necessary or there are value variation. It should be re-written. In fact, it is not discussed in Discussion section.

A: We agree with this remark. In our work, we refer to the currently observed edge computing trend, which is precisely aimed at analyzing data at the point of download to limit the amount of data transmitted via the network. In our opinion, this idea is not new, because such work is carried out. As a novelty, however, we found the algorithm itself adapted to this particular case. Our goal was to simplify the structure of the algorithm so that it translates into miniaturization of the entire system. One of our goals was also to make it robust against varying noise levels.

In the revised version of the manuscript, we have added a subsection (p.6.3), in which we conducted a complexity analysis of the circuit that implements the proposed algorithm. In general, we have mainly focused on simplifying the filters used. We used such filters, in which only summing and subtracting operations are used, instead of multiplication operations as in typical filers of this type.

R2: At the end of section 1, a paragraph about the structure of the contribution should be described

A: At the end of the introduction Section, we added a description of the structure of the article (lines 78-83).

R3: Section 1.1 should be numbered as 2. State of the art or Related work

A: We have changed the structure of the manuscript, accordingly.

R4: Section 2.2 is too long. It should be considered as a new section, for example Section 4

A: We have changed the structure of the manuscript, accordingly.

R5: The Conclusions should be rewritten because the reviewer finds them very generic.

A: Short conclusions section in the previous manuscript version was due to using by us the IMRaD approach. As we read the MDPI instruction, more important is the Discussion section, while the Conclusion one may even be omitted. Nevertheless, we extended it by addition a summary of the obtained results.

Round 2

Reviewer 2 Report

Dear Sponsor,

This reviewer understands that the author answered the requests.

Sucess

Reviewer 3 Report

The manuscript has been substantially improved in this revision. Thanks for the efforts, and I have no additional comments. 

Reviewer 4 Report

All comments have been taken in account by authors. The manuscript has been improved and now may be considered for publication in Remote Sensing journal.

However, the final decision belongs to the editor of the journal.

No further comments.

This manuscript is a resubmission of an earlier submission. The following is a list of the peer review reports and author responses from that submission.

Round 1

Reviewer 1 Report

Abstract is barely informative and needs great improvement

Keywords are too many and not really useful

Introduction (and related chapters) is not effective, unnecessarily far too long and the aim is not stated clearly, being rather vague

Methods are detailed enough, but the real predictive power of the whole system has remained quite obscure to me

In spite of the results, in the Discussion section I would have liked to see a possible application for indoor monitoring

Conclusions should be implemented since as such are just a short summary of the project

Author Response

Please find the response attached below.

Reviewer 2 Report

Overall, I think that the paper gives too much depth in describing in a simplified manner the air pollution monitoring systems, and too much general background on microsensors.  It has too little critical review of the literature to support the conclusions, and in particular, there is little discussion of the results of studies. I would encourage the authors to substantially strengthen the information regarding the steps fulfilled up to now regarding the microsensor’s development , which should be the focus of the paper.  If it is too difficult to do that for the wide scope that the paper currently has, then the authors might narrow the scope.  Furthermore, the introduction should contain a paragraph that describes the project that is suddenly presented in conclusions of the paper.

Second, the text should be checked for grammar and style. I recommend the support of a native English speaker or a professional proofing services.

Corresponding author does not have all the required information (e.g. phone number)

Abstract does not have conclusions and potential recommendations.

Introduction L12-19 does not have supporting references for the statements.

This is confusing:  In case of PM10 particles (diameter less than 10 mm) harmful are carcinogenic heavy metals:

17 benzopyrenes, furans, dioxins, etc. Replace colon with comma after heavy metals

Reformulate L23-24, L43-44, L198-199, L200-201, L264-265, L290-291

Reformulate “and other more deterministic data processing algorithms”

L161 typo FromFrom; again: „Am issue here is the complexity of this circuit”

L202-209 what about sensors’ resolution and compliance with the standards (e.g. standards for equivalent methods, such as CEN/TS 16450: 2013). For example, for PM10 monitoring, the EN12341 standard stipulates the gravimetric samplers (WRAC, HVS and LVS) with discs. Some analyzing methods proved consistency for measuring PM fractions such as BAM, TEOM or optical devices. However, the microsensors showed frequently instability in measuring PM. To ensure that microsensors and microstations have potential in providing concentrations close to the reality, protocols must be performed to demonstrate equivalency.  The authors should elaborate on these issues (accuracy and stability).

L238 The project was realized

prepared on the basis of Google maps – street view = using the Google maps – street view

L473 include the website in text

often exceeding acceptable standards = often exceeding the limit values.

It seems that figs 6 and 7 are identical. b) figure is missing.

Discussion – The discussion of available findings needs substantial improvement in organization; include limitations and future work. Furthermore, compare your approach with other reports from literature (pros and cons).

Conclusions. In my opinion, conclusions should be rewritten. The text regarding the project description does not present conclusions of the research.  

Author Response

Please find it attached below.

Reviewer 3 Report

General Comments and concerns:

The organization of this manuscript is not correct. The authors seem to be writing a journal and not a manuscript. For example, the introduction has no citations and is followed by another section that has citations. Section 1.1 is long and should be shortened. I encourage the authors to read how other papers are published in this journal [example latest pubs]:

https://www.mdpi.com/journal/remotesensing

Section 2.21 in Methods should be shortened. Please be concise and arrive at your point with fewer sentences possible. I got lost, what’s the name of low-cost sensors(s) used in this study? The manuscript has many grammatical errors. I recommend sending the manuscript to a native English speaker or a language reviewer.

Specific comments

Paragraph L12-L15: I have two comments 5 particles penetrate deeper into the long, depending on the size. Particles must be smaller than 1 um, nanoparticles, to enter the bloodstream. Please add citations for your statements. Paragraph L16-L19: Please add citations for your statements. I would keep L12-L15 as one paragraph that represents “the problem” or “why we care.” Paragraph L20-24: I have two comments Please add the PM WHO standards or your country's PM standards. Please re-write this sentence and do not end the sentence with the symbol % “Usually, such information is provided only after exceeding the norms by 24 several hundred %” Please re-write the sentence L26, “It is because especially road transport is one of the primary sources of air pollution in cities.” Because it is grammatically incorrect. I will not give other examples, and please send for language review. Please add titles (and units) to the y- and x-axis of Figure 5.

Author Response

(The authors gave the same response as above.)

Round 2

Reviewer 1 Report

this revised version is overall improved; I still have a concern for the introduction, which has remained really too long and requires to be shortened dramatically; minor issues:
keywords are not keyphrases, please rewrite
avoid using unintroduced acronyms in the abstract

Author Response

The reviewer has presented us with important comments and requests, which we believe have improved the quality of our manuscript significantly. We hope the revised manuscript is much closer to the high standard of Remote Sensing journal.

R1: This revised version is overall improved. 

A1: Thank you for this comment.

R2: I still have a concern for the introduction, which has remained really too long and requires to be shortened dramatically.

A2: We substantially shortened and rebuilt the Introduction section. However, we had to add several new paragraphs, as requested by another reviewer.

R3: Minor issues: keywords are not keyphrases, please rewrite avoid using unintroduced acronyms in the abstract.

A3: Keywords have been rewritten to be be less general. We also corrected the acronym issue.

Reviewer 2 Report

The authors managed to improve some points of the previous recommendations. However, there are two major issues that depreciate the value of the presented research:

English style and grammar (check the typos e.g. furants instead of furans). I recommend that each of 4 authors to read carefully the revised manuscript and after revisions to submit the manuscript to a native English speaker for style corrections. It is very important to increase the readability of the manuscript up to the required quality for publication in a high-impact international journal such as Remote Sensing. The Introduction still lacks the context of air pollutants effect on human health. I recommend the consultation of the following papers:

1. https://www.ncbi.nlm.nih.gov/pubmed/26697186

2. https://www.mdpi.com/2073-4433/9/4/150 

3. https://www.ncbi.nlm.nih.gov/pubmed/26744881

Afterwards, they can elaborate a comprehensive paragraph regarding the current problematics in the field. This will substantiate the necessity of developing new hardware and associated algorithms. 

Author Response

The reviewer has presented us with important comments and requests, which we believe have improved the quality of our manuscript significantly. We hope the revised manuscript is much closer to the high standard of Remote Sensing journal.

R1: The authors managed to improve some points of the previous recommendations. However, there are two major issues that depreciate the value of the presented research:
English style and grammar (check the typos e.g. furants instead of furans). I recommend that each of 4 authors to read carefully the revised manuscript and after revisions to submit the manuscript to a native English speaker for style corrections. It is very important to increase the readability of the manuscript up to the required quality for publication in a high-impact international journal such as Remote Sensing.

A1: After reading the text internally (in the team), we requested a commercial language correction of the article (the cerificate in attachment). We hope that the text is improved now.

R2: The Introduction still lacks the context of air pollutants effect on human health. I recommend the consultation of the following papers:
1. https://www.ncbi.nlm.nih.gov/pubmed/26697186
2. https://www.mdpi.com/2073-4433/9/4/150
3. https://www.ncbi.nlm.nih.gov/pubmed/26744881
Afterwards, they can elaborate a comprehensive paragraph regarding the current problematics in the field. This will substantiate the necessity of developing new hardware and associated algorithms.

A2: We have read the provided papers carefully. Based on them, we added a few paragraphs in the Introduction section (marked in green) outlining the problem and providing the background of the conducted investigations.

Reviewer 3 Report

The authors have addressed my comments. 

Author Response

The reviewer has presented us with important comments and requests, which we believe have improved the quality of our manuscript significantly. We hope the revised manuscript is much closer to the high standard of Remote Sensing journal.